# Magnetic field sensing for 2D fault localization in underground power cables

**Hamid Ali**[1]☉, **Aamir Qamar**[1]☉*, **Nayef Alqahtani**[2]☉, **Ali H. Alenezi**[3]☉, **Faheem Ali**[4]☉

1 Department of Electrical Engineering, COMSATS University Islamabad, Wah Campus, Wah Cantt, Pakistan, 2 Department of Electrical Engineering, College of Engineering, King Faisal University, Al-Hofuf, Al-Ahsa, Saudi Arabia, 3 Remote Sensing Unit, Electrical Engineering Department, Northern Border University, Arar, Saudi Arabia, 4 Department of Electrical Engineering, Faculty of Electrical and Computer Engineering, University of Engineering and Technology, Peshawar, Pakistan

☉ These authors contributed equally to this work.
* aamirqamar@ciitwah.edu.pk

**Data availability statement:** All relevant data are within the manuscript.

## Abstract

Localizing faults in underground cables is a two-dimensional inverse problem, including the depth as the vertical dimension and the horizontal distance from the supply point. In this paper, a new approach based on magnetic field sensing is developed to accurately locate faults in underground cables without physically accessing them. Three types of fault are addressed, line-to-ground fault (LGF), leakage current fault (LCF), and line-to-line fault (LLF). Based on magnetic field sensing, the cable is first localized. Afterwards, the depth is calculated using the Quotient method and finally, the horizontal distance is determined using Kirchhoff's law (KVL). The simulation results show average errors of 5.25%, 4.77%, and 3.53% for LGF, LCF, and LLF. respectively, while the experimental results yield errors of 20% and 29% for LGF and LLF. These findings highlight the reliability of the proposed approach. Additionally, since the method requires only a single magnetic field sensing device with minimal computational effort, it offers a practical and cost-effective solution for the location of underground cable faults.

## Introduction

The underground power cables (UPCs) are used in both power transmission and distribution systems. The UPCs are utilized particularly in metropolitan areas to provide power for industrial and residential consumers. They are usually laid in direct trenches below the earth's surface at depth ranges from 0.7 to 1.5m, depending on the design standards of the system [1]. When three single-core cables are laid down, a parallel distance of approximately 0.3m is kept among the cables to protect them from mutual heat [2]. The UPCs are considered stable and reliable yet they are prone to faults. The key factors catalyzing faults in cables include water trees, moisture, partial discharge (PD), mechanical stresses, and human errors during cable jointing and bending [3]. Common faults occurring in UPCs are LGF, LLF, and LCF.

Researchers over time conducted numerous studies and proposed various methods for fault localization in UPCs. In [4], the authors reviewed comparison of different impedance-based single-end and multi-end methods for symmetrical and unsymmetrical fault localization in underground power distribution systems. Among them, [5] proposed single end

**Funding:** The authors extend their appreciation to the Deanship of Scientific Research at Northern Border University, Arar, KSA for funding this research work through the project number "NBU-FFR-2025-2159-04". The funders had no role in study design, data collection and analysis, decision to publish, or preparation of the manuscript.

**Competing interests:** The authors have declared that no competing interests exist.

method, that provided effective results. However, it can only be applied to line-to-ground (LG) and three-phase-to-ground (LLLG) fault localization. In [6], an impedance-based approach measures current and voltage both at fault event for the classification and location of the fault in UPCs. Although, impedance-based methods need direct access to cables to measure the voltage and current for tracking fault positions. In addition, the depth and LCF of the cable are not considered. The authors in [7] presented a machine-learning approach that identifies the phase with a line to ground fault, fault's impedance level and the faulty segment of the cable. However, the credibility of the proposed method is lost due to extensive data analysis required, which demands high computational resources and does not accurately determine the fault location. In [8,9], non-invasive magnetic field sensing methodologies were presented. In these studies, the magnetic flux density of current-carrying cables was utilized to determine both the horizontal position and the vertical depth of the cable. However, these algorithms are unable to locate the fault point or its horizontal distance. Another study [10] initially detected the line to ground fault, and then the fault was localized by investigating the electromagnetic transient of the current signal. However, this approach is only applicable for LGF. In [11], the authors conducted the classification and localization of series and shunt faults for short length cables. The joint approach of Reflection Coefficient Spectrum and Matched Filter Matrix in the presence of noise is proposed. Nevertheless, the proposed method is laborious demanding qualified operators for investigating faults.

The study by [12] proposed time domain reflectometry (TDR) for identifying and localizing up to two partial discharges (PD) in underground cables. However, the study only focuses localizing PDs, and the system complexity increases while identifying multiple PDs in a cable. Another study in [13] presented a multi-ended and segmented correlation approach for locating the PDs in cables. The performance of the segmented correlation algorithm was found to be better than that of the multi-ended algorithm but this approach is not cost-effective due to numerous PD sensors required.

The Fourier transform is employed and the impedance of normal, with voids and short-circuited cables is calculated in [14]. The resulting magnitude and phase of frequency domain signals are analyzed. The signal information distinguishes the cable status. However, the ground faults are not considered.

In [15], internet of things (IoT) based sensors and in [16], and [17] distributed and micro phasor measurement units (PMUs) are deployed for detection and location of faults. Additionally, the authors in [18] optimized number of PMUs to observe electrical parameters for impedance-based fault localization algorithms in UPCs. However, time complexity, power consumption, and cost are high due to the large number of devices.

The cable insulation deteriorates over time causing aging issues and flunking the cable that impacts the reliability of the power system. The authors in [19] utilized the relative permittivity of cable to find the overall health of online cable. Although, this approach doesn't consider the LCF. In [20], the current measurement of Rogowski coil and in [21], the high-frequency PD measurement sensors are designed to monitor the cable condition and power quality online. However, the accuracy of sensors depends on the designed PD frequency, current range and voltage level of cable. The open circuit, short circuit, and leakage current faults positioning of the metallic sheath of high voltage cables for the cross bonded system are presented in [22,23]. Although, these approaches are applicable only for metallic sheath faults of high-voltage cables.

In this paper, a novel approach based on magnetic field sensing is proposed to locate underground cable faults. The proposed method localizes the cable fault by calculating both the cable's depth from the earth surface and the horizontal fault distance from the supply point. Initially, the cable is localized using the sensed magnetic field. Subsequently, the cable's

depth is calculated using the Quotient method that involves the ratio of the horizontal and vertical components of the measured magnetic flux density. Finally, the horizontal fault distance is calculated using KVL. The magnetic field required for this process can be measured using a Hall-effect sensor, which is suitable for low-frequency applications and can be repositioned across multiple measurement points. However, the proposed method is susceptible to surrounding electromagnetic interference (EMI), which can be mitigated using Blind Source Separation (BSS) approaches such as Independent Vector Analysis (IVA) and Independent Component Analysis (ICA), as discussed in [24–26].

The following contributions make the proposed approach superior to the existing methods:

- Non-invasive magnetic field sensing approach for fault localization in underground cables, eliminating the need for direct access to the cable
- The proposed method performs two-dimensional localization, simultaneously estimating both cable depth and horizontal fault distance, unlike prior magnetic sensing methods that only detect cable depth or fault distance
- Deployment of fewer sensors (a single sensing device moved across multiple measurement points, unlike existing approaches that rely on multiple fixed sensors)
- Cost effective

Acknowledging the real access to underground cables, lack of depth detection, limited ease of use, and the intensive computational demands of existing fault localization methods, the authors of this study emphasize the need for a new approach to locate the faults in UPCs without requiring physical access to cables while also calculating their burial depth and offering low computational costs. The approach would be capable to locate faults including determining the depth of UPCs. Such a method would greatly assist field operators by providing them with a single, accurate solution for fault localization in UPCs.

## Key research gaps

The identified gaps in the current literature underscore several important issues. For current measurement, the direct access to the cable is mandatory to install the instruments, which limits measurements to the installation site. Additionally, some methods are applicable to only one or two faults. None of the existing methods in the literature is compact, capable of locating multiple faults, and provides depth information at any point along the cable through virtual access. Some proposed techniques require an excessive number of measurements and sensor deployments, which necessitate an established system to gather and process the sensors information. This increases the cost and processing time of the system and undermines the credibility and practicality of these methods.

## Problem statement

Localizing faults in the underground power cables without direct access and in a cost-effective manner is essential. Moreover, this is a multidimensional inverse problem, the depth of UPCs is one of the components to locate faults that remains unknown due to back-fill during maintenance. In addition, the lack of depth information for UPCs poses risks to excavation workers during maintenance, highlighting the need for accurate depth detection for UPCs. Therefore, it is necessary to develop a method to identify the fault points and cable depth simultaneously from the ground surface by virtually accessing UPCs, utilizing minimal equipment and personnel from power utilities.

## Mathematical modeling

In this section, a mathematical expression is developed using the proposed approach to locate the fault point in underground power cables. Initially, the fault point location expression is developed for the LGF, followed by LCF and LLF.

### LGF

Fig 1 illustrates a single core cable of length $L$ with line-to-ground fault point at $F_{LG}$. $X$ represents the distance of $F_{LG}$ from the supply terminal while L-X is the length of the remaining cable from the fault point. Additionally, $R_X$ shows the resistance of the cable up to the fault point, $R_g$ is the ground resistance, and $I_F$ is the fault current flowing in the cable as a DC supply V is applied to the cable. According to the Kirchhoff's law, the applied voltage V is expressed as:

$$V = I_F R_X + I_F R_g \qquad (1)$$

$$X = \frac{A}{\rho I_F}(V - I_F R_g) \qquad (2)$$

The $\rho$ and $A$ are the material resistivity and cross-sectional area of the cable's conductor. Their values are known as provided by the manufacturer. The ground resistance $R_g$ varies with fault conditions, however, its standard value usually lies between 0.1 and 5$\Omega$ [27]. Since, $R_g$ can be measured using well established techniques such as Fall-of-potential and the Wenner method, it is assumed constant in this study to focus on the applicability of the proposed method. The $I_F$ is determined based on magnetic field sensing on the earth's surface without direct access to the cable. $M$ in Fig 1 illustrates the magnetic field sensing line on the earth's surface.

To calculate the fault current $I_F$ flowing in the cable, it is assumed that the cable is stationed along the x-axis which is illustrated in Fig 2. This cable carries a fault current $I_F$ and is buried in the soil at depth $h$ from the ground surface. According to Ampere's law, the magnetic flux density at point P and at a distance $r$ from the cable at the ground surface is $\vec{B}_\phi$,

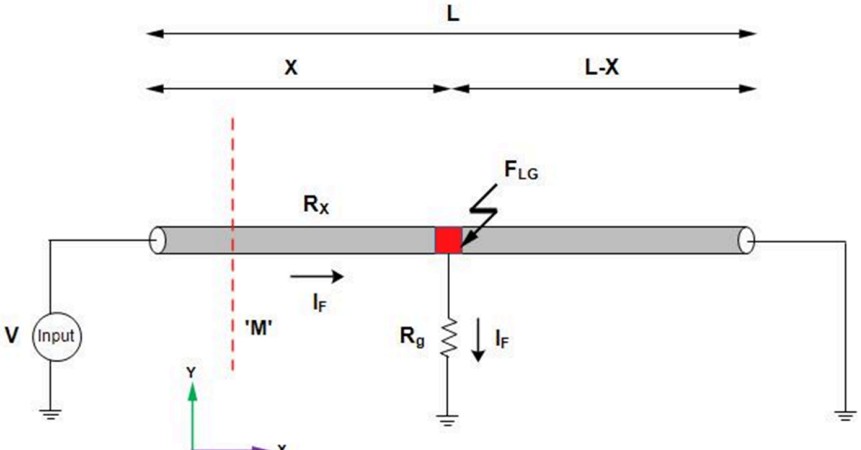

**Fig 1. LGF in a single core underground cable.**

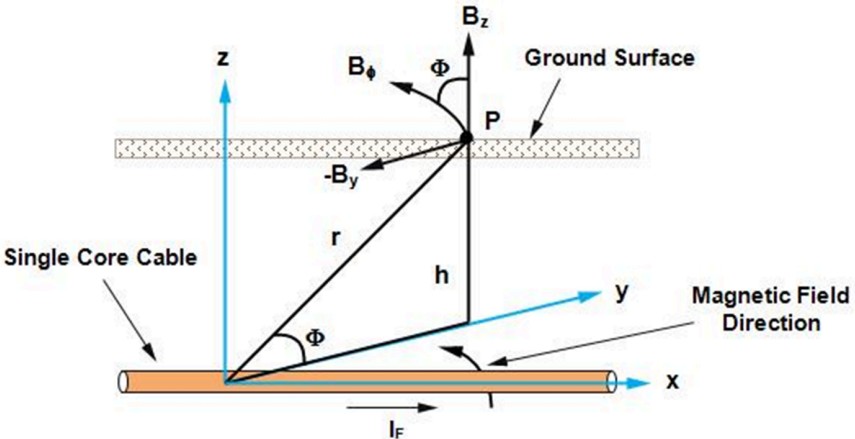

**Fig 2. A single core cable stationed along the x-axis is buried at a depth of *h* below the earth's surface.** The fault current $I_F$ produces a magnetic flux density $\vec{B}_\phi$, which is measured at point *P*. The *r* represents the distance between the cable and point *P* and $\phi$ is the angle between $\vec{B}_\phi$ and $\vec{B}_z$ components.

which is expressed as:

$$\vec{B}_\phi = \frac{\mu I_F}{2\pi r}\hat{a}_\phi \tag{3}$$

The $\mu$ is the permeability of soil and $\hat{a}_\phi$ is the direction of magnetic flux density circulating around the cable. Expressing $\vec{B}_\phi$ in rectangular coordinates, the horizontal and vertical components $\vec{B}_y$ and $\vec{B}_z$ are expressed as:

$$\vec{B}_\phi = \frac{\mu I_F}{2\pi r}\left(-sin\phi\hat{a}_y + cos\phi\hat{a}_z\right) \tag{4}$$

$$\vec{B}_y = \frac{-\mu I_F h}{2\pi\left(y^2 + h^2\right)}\hat{a}_y \tag{5}$$

$$\vec{B}_z = \frac{\mu I_F y}{2\pi\left(y^2 + h^2\right)}\hat{a}_z \tag{6}$$

The $\vec{B}_y$ at y = 0m is maximum, corresponding to the location of fault under consideration as shown in Fig 1. Consequently, (5) simplifies to:

$$\left|\lim_{y\to 0}\vec{B}_y\right| = \lim_{y\to 0}\frac{\mu I_F h}{2\pi\left(y^2 + h^2\right)}\hat{a}_y \tag{7}$$

$$I_F = \frac{2\pi h\left|\lim_{y\to 0}\vec{B}_y\right|}{\mu} \tag{8}$$

The $I_F$ is the fault current flowing in the cable and *h* is the depth of the cable from the earth's surface that is unknown due to back-fill and maintenance. Therefore, the *h* is calculated as in [28] based on the Quotient method which evaluates the ratio of the horizontal

and vertical components of magnetic flux density. Considering (5) and (6), the $h$ based on Quotient method is expressed as:

$$h = y \frac{|\vec{B_y}|}{|\vec{B_z}|} \qquad y \neq 0 \tag{9}$$

The $y$ represents the magnetic flux density sensing point along the magnetic field sensing line $M$.

## LCF

Voids, cracks, moisture, and other chemical reactions are the main culprits responsible for the weakening of cable insulation. This degradation enables a small but continuous leakage current to flow from the conductor to the ground through the compromised insulation without forming a complete short-circuit, unlike in the case of an LGF. Persistent leakage current in underground cables progressively deteriorates the insulation material, which may ultimately lead to insulation breakdown and evolve into a major fault condition if not mitigated [29,30].

To locate the location of LCF, Fig 3 illustrates a single core cable of length $L$ with LCF at point $F_{LC}$. This $F_{LC}$ is at a distance $X$ from the supply terminal while $L$–$X$ shows the length of the remaining cable from the point $F_{LC}$. Applying a DC voltage $V$ across the cable, $I_1$ is the current flowing from the supply terminal to $F_{LC}$. $I_2$ is the current flowing from $F_{LC}$ in the remaining cable, $I_L$ is the leakage current leaking in the ground, and $V_F$ shows the potential at $F_{LC}$. $R_L$ illustrates the total resistance of the cable, $R_X$ is the resistance of the cable from the supply terminal to the fault point, $R'_X$ is the resistance of the remaining cable from the fault point, $R_{LC}$ appears for the leakage resistance, and $R_g$ represents the ground resistance. The current $I_1$ and $I_2$ are calculated based on magnetic field sensing by measuring the magnetic flux densities along lines $M_1$ and $M_2$. The methodology is discussed in detail in Section LGF.

Using nodal analysis at the point $F_{LC}$ in Fig 3, the potential $V_F$ at $F_{LC}$ is expressed as:

$$V_F = V - I_1 R_X \tag{10}$$

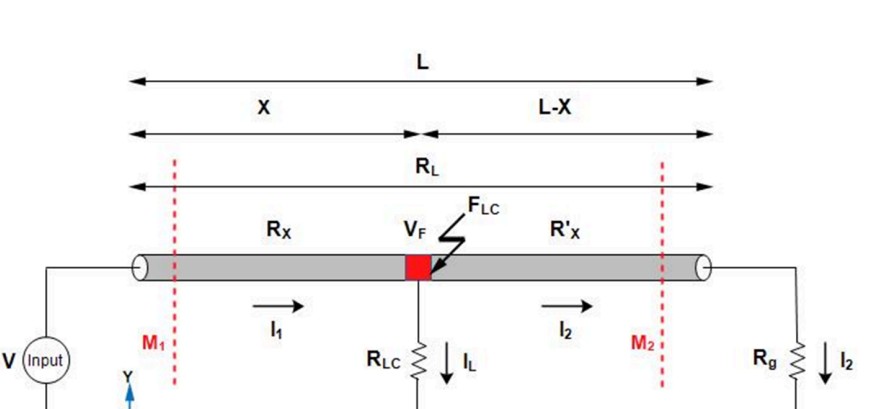

**Fig 3. LCF in a single core underground cable.**

and

$$V_F = I_2(R'_X + R_g) \tag{11}$$

Using (10) and (11),

$$R_X = \frac{V - I_2(R_g + R_L)}{I_L} \tag{12}$$

where $R_L = R_X + R'_X$

$$X = \frac{A(V - I_2(R_g + R_L))}{\rho I_L} \tag{13}$$

The $X$ shows the location of $F_{LC}$ from the supply terminal.

## LLF

The line-to-line fault point, represented as $F_{LL}$ and illustrated in Fig 4, is located at a distance $X$ from the supply terminal. Similarly, the $L–X$ shows the length of the remaining cables from the fault point. To calculate the location of the LLF, a DC potential V is applied across the short-circuited cables of length $L$, which are separated by a distance $d$. This causes the fault current $I_F$ to drift in the cables from high potential towards low potential. In Fig 4, $R_X$ represents the resistance of the cable from the supply terminal to the fault point $F_{LL}$. Additionally, line $M$ represents the magnetic field sensing line. It must be noted that during the LLF, the net cable resistance observed by the fault current $I_F$ is $2R_X$. Therefore, the applied potential is expressed as:

$$V = 2I_F R_X \tag{14}$$

$$X = \frac{VA}{2\rho I_F} \tag{15}$$

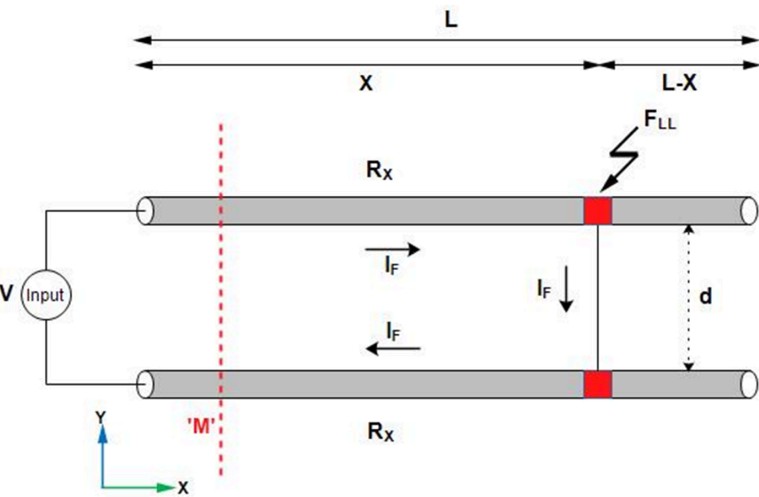

**Fig 4. LLF in single core underground cables.**

The $I_F$ is calculated on the earth's surface based on magnetic field sensing without direct access to the cables. Fig 5 illustrates the fault current calculation using magnetic field sensing.

The short-circuited cables illustrated in Fig 4 are considered in Fig 5 to calculate the fault current on the earth's surface without accessing the cables. It is assumed that the cables are positioned along the x-axis such that the current $I_F$ flows in the positive x-direction in the cable located at $y = d/2$. Similarly, in the cable located at $y = -d/2$, the direction of $I_F$ is reversed. As a consequence of current flow, the magnetic flux density is generated and travels to the earth's surface. The $\vec{B}_{\phi_1}$ is the magnetic flux density due to the cable at $y = d/2$ and $\vec{B}_{\phi_2}$ is the magnetic flux density due to cable at $y = -d/2$. According to Fig 5, $h$ represents the depth of cables, $r_1$ and $r_2$ represent the distances of the cables from the point $P$ on the earth's surface, and $d$ shows the separating distance between the two cables. Additionally, $\phi_1$ is the angle between $\vec{B}_{\phi_1}$ and its component $-\vec{B}_{y_1}$, and $\phi_2$ is the angle between $\vec{B}_{\phi_2}$ and its component $-\vec{B}_{z_2}$. The magnetic flux density from the two cables is expressed as:

$$\vec{B}_\phi = \vec{B}_{\phi_1} + \vec{B}_{\phi_2} \tag{16}$$

The $\vec{B}_\phi$ in rectangular coordinates is expressed below:

$$\vec{B}_\phi = \frac{\mu I_F}{2\pi r_1}\left(-sin\phi_1\hat{a}_y + cos\phi_1\hat{a}_z\right) + \frac{\mu I_F}{2\pi r_2}\left(sin\phi_2\hat{a}_y - cos\phi_2\hat{a}_z\right) \tag{17}$$

Similarly, the horizontal component $\vec{B}_y$ and vertical component $\vec{B}_z$ associated with $\vec{B}_\phi$ are expressed as:

$$\vec{B}_y = \frac{\mu I_F}{2\pi}\left[\frac{h}{(y + d/2)^2 + h^2} - \frac{h}{(y - d/2)^2 + h^2}\right]\hat{a}_y \tag{18}$$

$$\vec{B}_z = \frac{\mu I_F}{2\pi}\left[\frac{(y - d/2)}{(y - d/2)^2 + h^2} - \frac{(y + d/2)}{(y + d/2)^2 + h^2}\right]\hat{a}_z \tag{19}$$

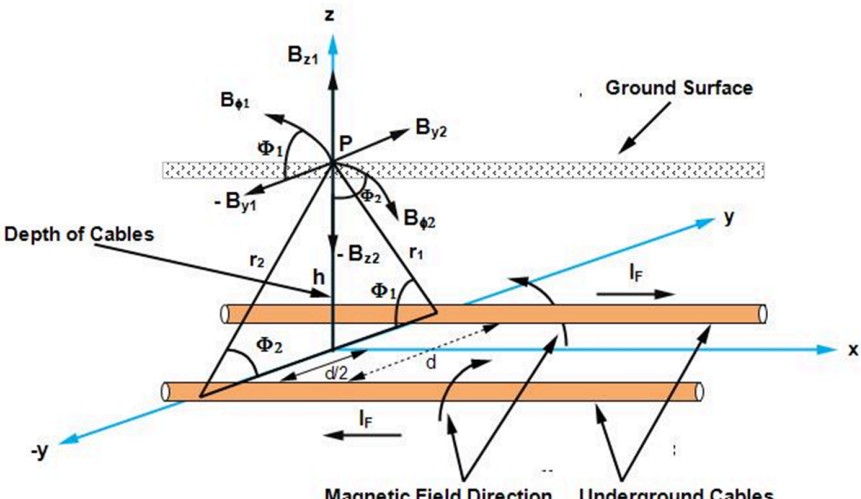

**Fig 5. Magnetic flux density at point *P* on the earth's surface from two single core, short-circuited cables carrying $I_F$ as the fault current.** The cables are separated by a distance $d$ and buried at a depth $h$ below the earth's surface.

It can be observed from (19), that the $\vec{B}_z$ is maximum at y = 0m, representing the center or location of the short-circuited cables. Furthermore, using (19), the fault current $I_F$ is expressed as:

$$I_F = \frac{2\pi \left| \lim_{y \to 0} \vec{B}_z \right|}{\mu \left[ \frac{d}{(d/2)^2 + h^2} \right]} \tag{20}$$

The $d$ is the separation distance between the two cables, $\mu$ is the permeability of soil, and $h$ is the depth of the cables from the earth's surface. Here the $h$ is unknown, that is calculated using the Quotient method [28]. Using (18) and (19), at $y = d/2m$, the depth $h$ is expressed as:

$$h = d \frac{|\vec{B}_z|}{|\vec{B}_y|} \tag{21}$$

## Results and discussion

This section illustrates the simulations conducted to validate the viability of the proposed method for fault localization in the underground cables. The simulations, performed in COMSOL Multiphysics 6.1 [31], aim to evaluate the fault location. The simulations are conducted in COMSOL's AC/DC module using magnetic and electric field physics (MEF). Additionally, throughout the simulations the effects of temperature and electric fields from the underground cables are ignored.

### LGF

To demonstrate the viability of the proposed method for the line-to-ground fault localization in single core underground cable, Fig 6 shows the simulation environment developed in COMSOL Multiphysics 6.1. A single core cable with a length of 1000m is positioned along the x-axis and buried at a depth of 1m from the earth's surface. The cable conductor is made of copper with a conductivity of $5.998 \times 10^7 \text{S/m}$, relative permeability 1, and a cross-sectional area $1.256 \times 10^{-3} \text{m}^2$. Additionally, the PVC insulation layer is 0.03m thick and the grounding resistance is assumed to be 0.294Ω. Based on magnetic field sensing, a 1000V DC voltage is applied to measure the magnetic flux density on line M at the earth's surface. Line M spans from -2m to 2m along the y-axis and obtained using the Cut Line 3D feature within the 'Datasets' node of COMSOL. The geometry is meshed using user-controlled meshing, with the global meshing size set to extra fine and the local meshing sizes set to extremely fine, using edge, free triangle and free tetrahedral elements. Moreover, the system took approximately 7 hours to compute the model.

Initially, the LGF represented by $F_a$ at 100m from the supply point is considered as illustrated in Fig 6. In order to locate $F_a$, a 1000V DC is supplied and the magnetic flux density generated is sensed on line $M$, which is illustrated in Fig 7. It is observed that the $|\vec{B}_y|$ is maximum at y = 0m, which indicates the location of cable under fault consideration. Additionally, the surface plot of the magnetic flux density distribution (MFDD) is shown in Fig 8, which identifies LGF point up to 100m. It is evident from the figure that the MFDD is constant up to the fault point, due to the constant current flow in the cable. However, beyond the fault point, the current becomes zero and the MFD also drops to zero, indicating the short circuit point of the cable with the earth. The leakage current of the remaining cable segment is disregarded.

In order to locate $F_a$, the depth of the cable is calculated based on Quotient method using (9). It is illustrated in Fig 7 by the red dotted graphs, the $|\vec{B}_y| = 5.195 \times 10^{-4} \text{T}$ and $|\vec{B}_z| = 2.646 \times 10^{-4} \text{T}$ at y = 0.5m. According to (9), the depth is calculated as 0.98m with a percentage error of 2%. The depth calculations are tabulated in Table 1. Similarly, for the fault distance,

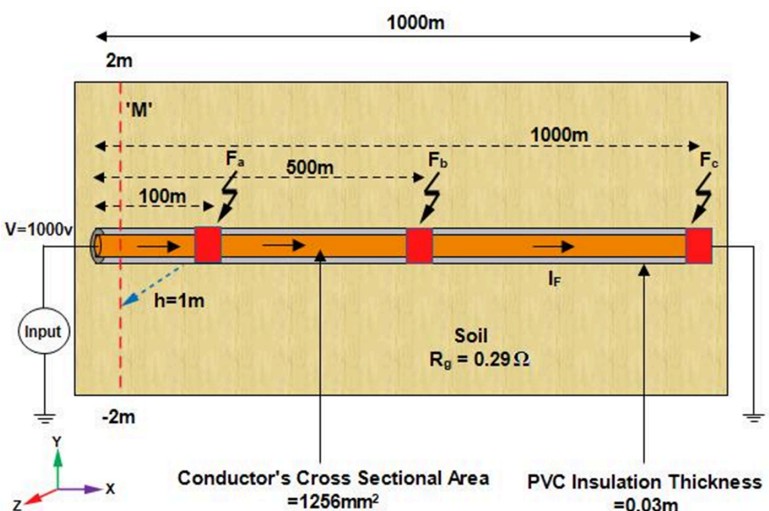

**Fig 6. The simulation environment illustrates a single core cable positioned along the x-axis and buried at depth h = 1m from the earth's surface.** Line to ground fault at 100m, 500m, and 1000m along the length of the cable is considered. The cable carries $I_F$ as the fault current and M represents the magnetic field measurement line.

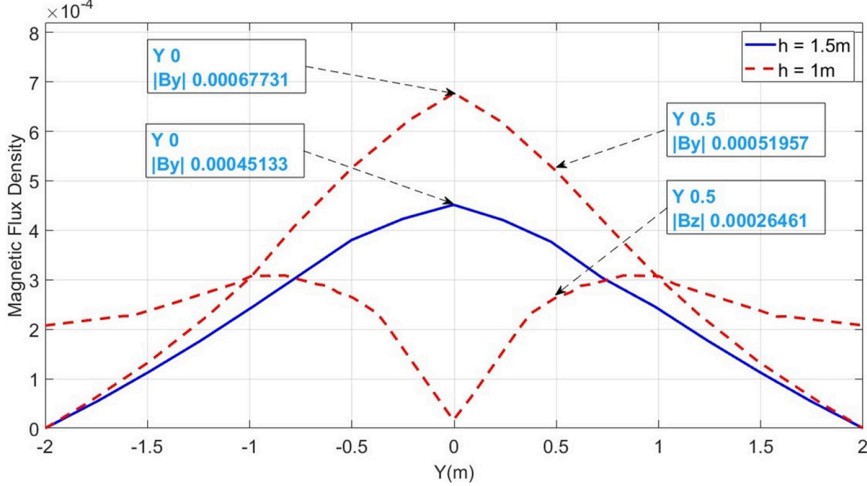

**Fig 7. $\vec{B}_y$ and $\vec{B}_z$ measured along the line M for LGF at 100m.** The red dotted graphs represent magnetic flux densities when the cable is buried at h = 1m while the blue graph illustrates $\vec{B}_y$ when the cable is buried at h = 1.5m.

the initial fault current $I_F$ is calculated using (8), which is equal to 3386.5A and finally, the fault distance is determined based on (2). The fault distance calculated equals 96.48m with a percentage error of 3.52%.

To further validate the proposed method, the cable burial depth was changed to h = 1.5m. The $|\vec{B}_y|$ sensed on line M is shown in Fig 7 as the blue graph. It is observed that $|\vec{B}_y|$ equals $4.5133 \times 10^{-4}$T at y = 0m and the calculated fault distance equals 92.21m, resulting the percentage error of 7.78%.

To further concrete the validity of the proposed method, the fault points $F_b$ at 500m and $F_c$ at 1000m are considered, as illustrated in Fig 6. The magnetic flux density $|\vec{B}_y|$ sensed on

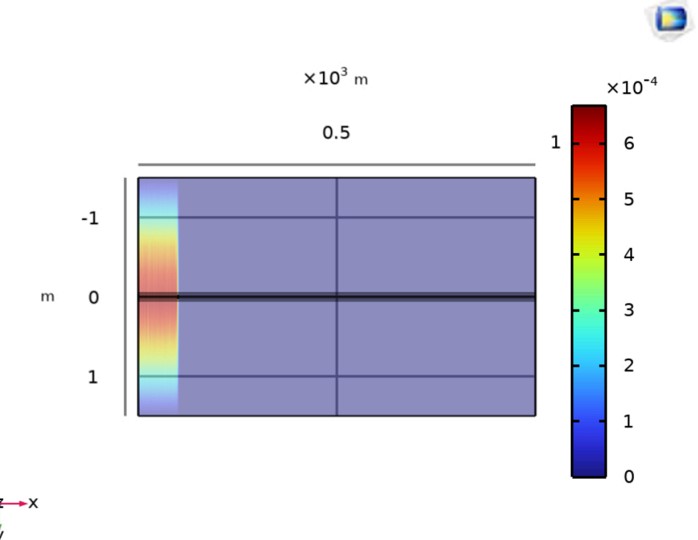

**Fig 8. Distribution of $\vec{B}_y$ on the earth's surface for LGF at 100m with the cable buried at h = 1m.**

**Table 1**. **Depth calculated using the Quotient Method for h = 1m, considering LGF, LCF, and LLF.**

| Fault | y(m) | $|\vec{B}_y|$(mT) | $|\vec{B}_z|$(mT) | $h_{cal}$(m) | E(%) |
|---|---|---|---|---|---|
| LGF | 0.5 | 0.51957 | 0.26461 | 0.98 | 2 |
| LCF | 1 | 7.91 | 7.93 | 0.997 | 0.25 |
| LLF | 0.15 | 5.39 | 17.787 | 0.989 | 1.1 |

line M for $F_b$ and $F_c$ is shown in Figs 9 and 10. Additionally, Figs 9 and 10 also illustrate the $|\vec{B}_y|$ for $F_b$ and $F_c$ when the cable depth was changed from h = 1m to h = 1.5m. The calculated current $I_F$, fault distance $X_{cal}$ and percentage errors for $F_a$, $F_b$, and $F_c$ are summarized in Table 2. It is observed from Table 2 that the average percentage error for $F_a$, $F_b$, and $F_c$ is 5.25%. Moreover, The surface plots of $|\vec{B}_y|$ distribution are shown in Figs 11 and 12.

COMSOL Multiphysics employs the finite element method (FEM) as its underlying computational technique. To ensure the accuracy and reliability of the FEM-based tool, the model with the fault point at 1000m and the cable buried at h =1m was subjected to a mesh-independence test using different discretizations levels [32]. The convergence of the results with mesh refinement confirms the stability of the numerical solution. A detailed summary of the mesh-independence tests from fine to extra fine meshes is provided as follows.

Methodology:

Three distinct configurations are analyzed through simulation:

Fine Configuration:

1595520 Tetrahedra elements, 155253 Triangle elements, 65472 edge elements, 32 vertex elements, and 10,579,045 degrees of freedom (DOF)

Minimum Element Size: 0.1m

Maximum Element Size: 80.2m

Finer Configuration:

2048805 Tetrahedra elements, 156996 Triangle elements, 65502 edge elements, 32 vertex elements, and 13,455,945 DOF

Minimum Element Size: 0.1m

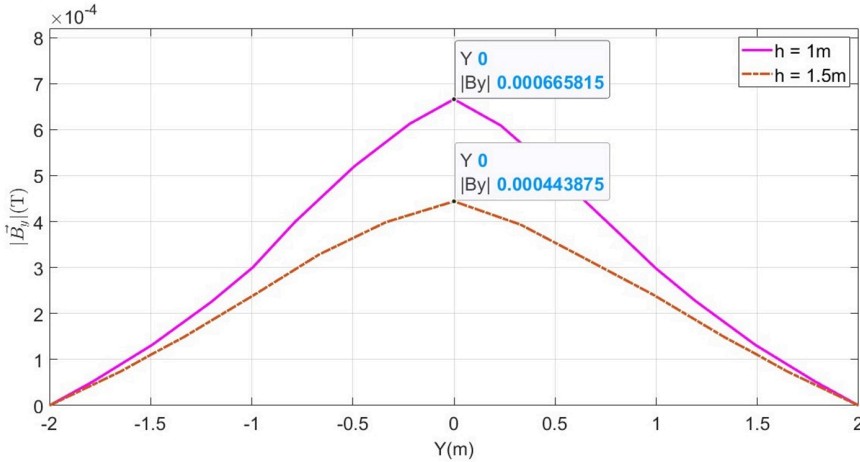

**Fig 9. Distribution of $\vec{B}_y$ along line M for LGF at 500m when the cable is buried at h = 1m and h = 1.5m.**

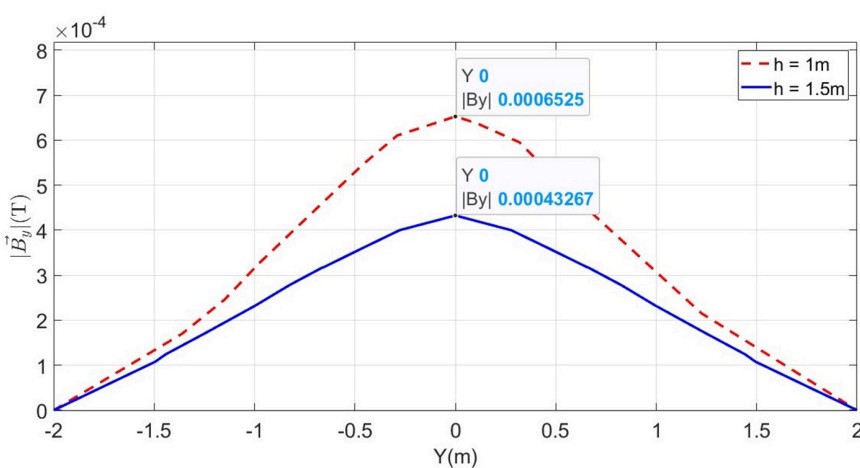

**Fig 10. Distribution of $\vec{B}_y$ along line M for LGF at 1000m when the cable is buried at h = 1m and h = 1.5m.**

**Table 2. Outcomes of line to ground faults at 100m, 500m, and 1000m.**

| $h$(m) | $|\vec{B}_y|(\mu T)$ | $I_F$(A) | $X_{act}$(m) | $X_{cal}$(m) | $E_\%$ | $R_g(\Omega)$ |
|---|---|---|---|---|---|---|
| 1 | 677.31 | 3386.5 | 100 | 96.48 | 3.51 | 0.29401 |
| 1.5 | 451.333 | 3387.15 | 100 | 92.21 | 7.78 | 0.29401 |
| 1 | 665.815 | 3329.07 | 500 | 480.8 | 3.91 | 0.29401 |
| 1.5 | 443.875 | 3329.06 | 500 | 480.49 | 3.9 | 0.29401 |
| 1 | 652.5 | 3262 | 1000 | 942.55 | 5.74 | 0.29401 |
| 1.5 | 432.67 | 3245.02 | 1000 | 1058.5 | 6.68 | 0.29401 |

Maximum Element Size: 80.2m

Extra fine Configuration:

2607738 Tetrahedra elements, 156996 Triangle elements, 65502 edge elements, 32 vertex elements, and 17,006,511 DOF

Minimum Element Size: 0.1m

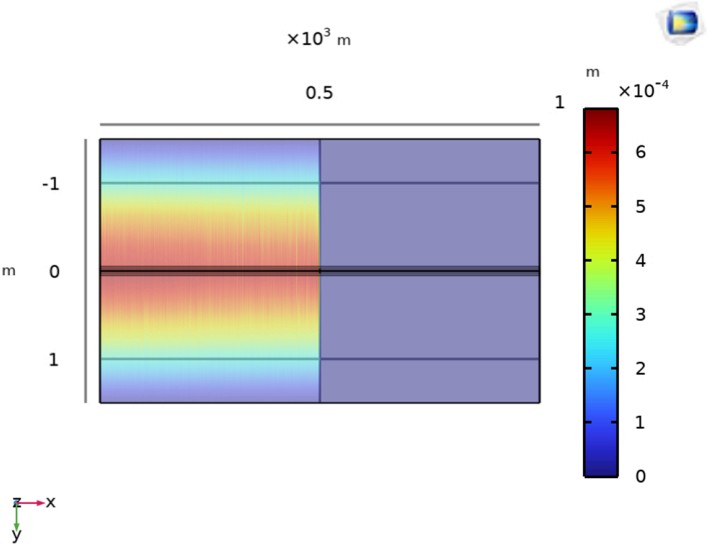

**Fig 11. Distribution of $\vec{B}_y$ at the earth's surface for LGF at 500m with the cable buried at h = 1m.**

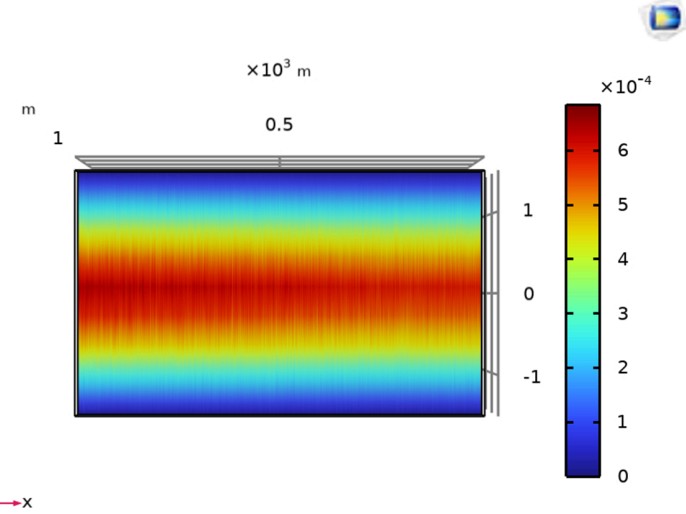

**Fig 12. Distribution of $\vec{B}_y$ at the earth's surface for LGF at 1000m with the cable buried at h = 1m.**

Maximum Element Size: 55m
Computational Time:
Fine Configuration: 1h 04min
Finer Configuration: 4h 35min
Extra Fine Configuration: 7h 00min
Solver Configuration:
Linear Solver
Type: Iterative
Solver: FGMRES (Flexible Generalized Minimal Residual)
Maximum Iterations: 10,000

Non-linear Solver:

Method: Newton (Automatic)

Maximum Non-linear Iterations: 25

Termination Criterion: Solution or residual

Residual Factor: 1000

System Hardware Specifications:

Processor: 12th Gen Intel(R) Core(TM) i7-12700k

RAM: 32GB DDR5

Storage: Lexar SSD NM620 1TB

Results:

By comparing the fine, finer and extra fine configurations, the calculated depths were 1.26 m (26% error), 1.16 m (16% error), and 0.98 m (2% error), while the corresponding fault distances were 1767.5 m (76.8% error), 1393.6 m (39.4% error), and 942.55 m (5.7% error) respectively, confirming that the extra-fine mesh yields the most accurate results relative to the actual depth (1 m) and fault distance (1000 m). However, this improvement comes at the cost of a significantly larger number of elements, which in turn increases the overall computational time and requires greater memory allocation.

## LCF

The feasibility of the proposed method for LCF localization is validated using the same simulation environment as shown in Fig 6. A single core cable buried at h = 1m and with a length of 1000m is positioned along the x-axis with identical parameters with those of the LGF model, including conductor's material, cross sectional area, conductivity, cable depth, etc. The grounding resistance is assumed to be 0.294534$\Omega$. A 1000V DC is applied to measure the magnetic flux density on lines $M_1$ and $M_2$ at the earth's surface as depicted in Fig 3. Both lines $M_1$ and $M_2$ extend along the y-axis from -2m to 2m. The geometry is finely meshed using user-controlled meshing. Furthermore, the system took approximately 3 hours to compute the model.

Using the proposed method to localize LCF, initially, $F_a$ is considered at 100m. The magnetic flux density $\vec{B}_{y_1}$ sensed along line $M_1$ and $\vec{B}_{y_2}$ along line $M_2$ is illustrated in Fig 13 by the red and pink dotted graphs. According to Fig 3, the $I_2$ is less than $I_1$ as a result of leakage current $I_L$ into the ground. This effect is vivid from the magnitudes of $\vec{B}_{y_1}$ and $\vec{B}_{y_2}$ in Fig 13.

According to (9), the depth of the buried cable is found to be 0.997m, using the ratio of $\vec{B}_{y_1}$ and $\vec{B}_{z_1}$ at $y = 1m$. The details of depth parameters are tabulated in Table 1. Moreover, at y = 0m, the peak value of $\vec{B}_{y_1}$ is 1.6626×$10^{-4}$T and $\vec{B}_{y_2}$ is 1.6243×$10^{-4}$T. Using these values and (8), $I_1$ and $I_2$ are calculated as 3325.2A and 3248.6A, respectively, while $I_L$ is calculated using Kirchhoff's current law to be 76.6A. Subsequently, the fault distance is calculated using (13) to be 98.8m.

To further test the viability of the proposed method, the buried depth of the cable was changed to h = 1.5m. The magnetic flux densities $\vec{B}_{y_1}$ and $\vec{B}_{y_2}$ sensed along lines $M_1$ and $M_2$, respectively, are illustrated by the blue and mustard colored dotted graphs in Fig 13. It is shown that $\vec{B}_{y_1}$ is 1.1084 × $10^{-4}$T and $\vec{B}_{y_2}$ is 1.1082 × $10^{-4}$T. The fault distance is calculated to be 104.83m.

Additionally, the proposed method is also tested for the LCF point $F_b$ at 500m for the cable buried at a depth of h = 1m and h = 1.5m. The measured $\vec{B}_{y_1}$ and $\vec{B}_{y_2}$ along lines $M_1$ and $M_2$ are shown in Fig 14. The fault distances calculated for h = 1m and h = 1.5m are 532.54m and 531.83m, respectively. The complete results for LCF are summarized in Table 3.

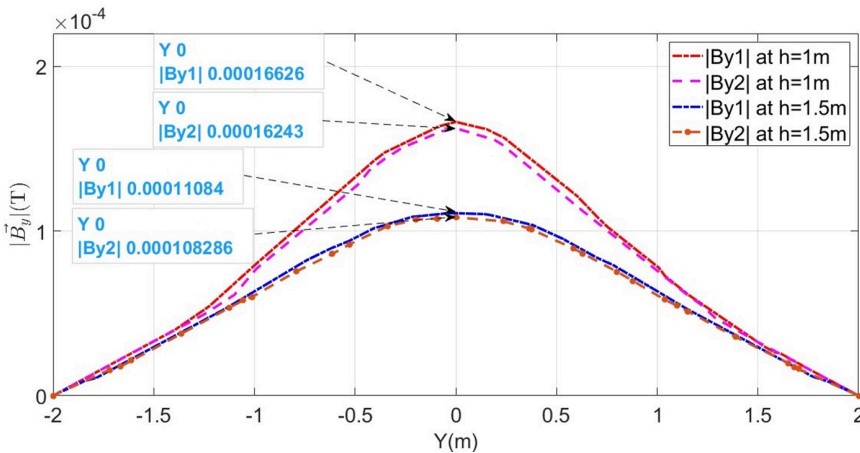

**Fig 13. Distribution of $\vec{B}_{y_1}$ and $\vec{B}_{y_2}$ along lines M$_1$ and M$_2$ for LCF at 100m when the cable is buried at h = 1m and h = 1.5m.**

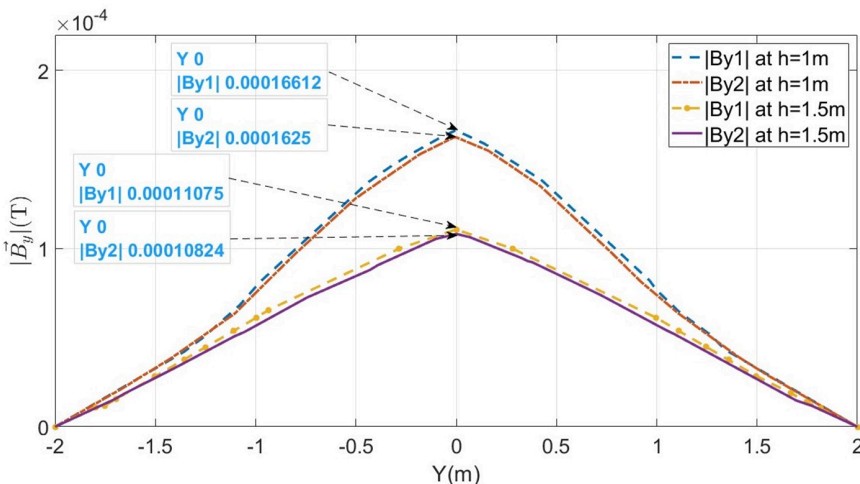

**Fig 14. Distribution of $\vec{B}_{y_1}$ and $\vec{B}_{y_2}$ along lines M$_1$ and M$_2$ for LCF at 500mm when the cable is buried at h = 1m and h = 1.5m.**

**Table 3. Outcomes of leakage current faults (LCF) at 100m and 500m.**

| $h$(m) | $R_g(\Omega)$ | $|\vec{B_{y1}}|(\mu T)$ | $I_1$(A) | $|\vec{B_{y2}}|(\mu T)$ | $I_2$(A) | $I_L$(A) | $X_{act}$(m) | $X_{cal}$(m) | $E_\%$ |
|---|---|---|---|---|---|---|---|---|---|
| 1 | 0.294534 | 166.26 | 3325.2 | 162.43 | 3248.6 | 76.6 | 100 | 98.8 | 1.19 |
| 1.5 | 0.294534 | 110.84 | 3325.2 | 110.829 | 3248.58 | 76.62 | 100 | 104.83 | 4.83 |
| 1 | 0.294534 | 166.12 | 3322.4 | 162.36 | 3247.2 | 75.2 | 500 | 532.54 | 6.5 |
| 1.5 | 0.294534 | 110.75 | 3322.5 | 108.24 | 3247.2 | 75.3 | 500 | 531.83 | 6.36 |

## LLF

Two single core short-circuited cables, separated by a distance of d = 0.3m, positioned along the x-axis and buried at a depth of h = 1m from the earth's surface, as illustrated in Fig 15, are considered to demonstrate the viability of the proposed method for LLF localization. The cables are 1000m long and their conductors are made of copper with a conductivity of

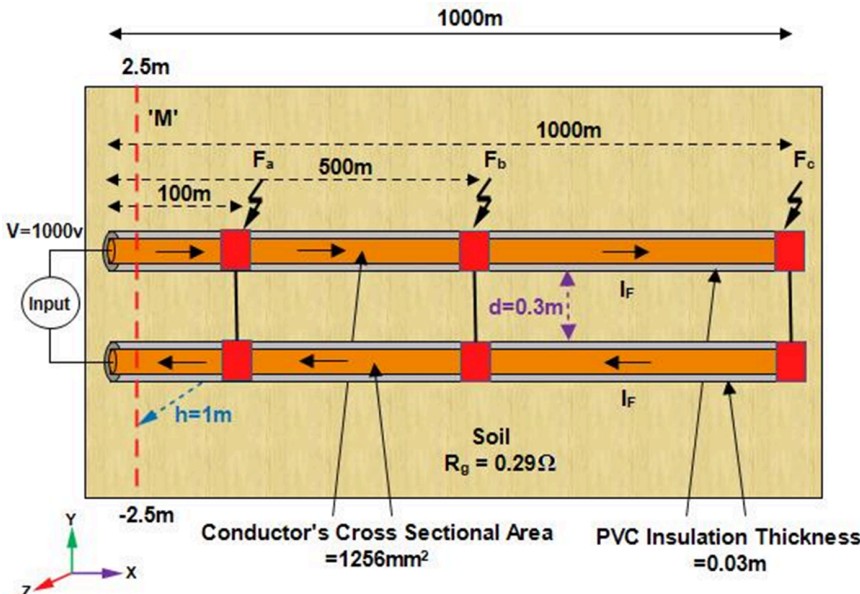

**Fig 15. Two single core short-circuited cables with line to line fault.** The cables are stationed along the x-axis and buried at depth h = 1m, maintaining a separation of d = 0.3m. LLF at 100m, 500m, and 1000m are shown by $F_a$, $F_b$, and $F_c$.

$5.998 \times 10^7$S/m and relative permeability of 1. The radii of the conductors are 0.02m. Additionally, the PVC insulation layer of the cables is 0.03m thick. A 1000V DC is applied across the cables, causing the $I_F$ to flow. The emanating magnetic flux density is sensed along line M at the earth's surface, as illustrated in Fig 15. Line M spans from –2.5m to 2.5m along the y-axis. The geometry is meshed using user-controlled meshing, keeping the minimum element size at 0.154m and maximum element size at 20.4m. Additionally, the system took approximately 7 hours and 15 minutes to compute the model.

The $F_a$ representing LLF at 100m from the supply point, is initially considered, as illustrated in Fig 15. The magnetic flux densities $\vec{B}_y$ and $\vec{B}_z$ sensed along line $M$ are illustrated in Fig 16 with blue graphs. It is observed that the $\vec{B}_z$ is maximum at y = 0m, showing the location of short- circuited cables. This is due to the integration of both cables magnetic fields, which represents the mid-point of cables under fault consideration. Moreover, from the figure, the value of $\vec{B}_y$ at y = 0.15m is $5.39 \times 10^{-3}$T and that of $\vec{B}_z$ at the same point is $17.78 \times 10^{-3}$T. Therefore, according to (21), the depth of the cables is found to be 0.989m with a percentage error of 1.1%.

The horizontal fault distance is calculated using (15). Fig 16 shows that $\vec{B}_z$ is equal $21.65 \times 10^{-3}$T at y = 0m. Therefore, according to (20), the fault current $I_F$ is calculated to be $368.95 \times 10^3$A. Based on (15), the fault distance is 102.14m.

The surface plot of the MFDD for LLF $F_a$ is shown in Fig 17. The $\vec{B}_z$ is detected up to the fault point (100m) due to the flow of current in the cables. However, beyond the fault point, the MFDD ceases due to the circuit discontinuity in the cables. Additionally, the MFDD is dark red in the middle of cables as shown by the color scale in the figure, indicating the maximum magnitude of $\vec{B}_z$. The leakage current in the remaining part of the cable is ignored. Verifying the proposed method for LLF localization at different depths of the short-circuited cables, the cables depth is adjusted to h = 1.5m. The $\vec{B}_z$ measured along line M is shown in

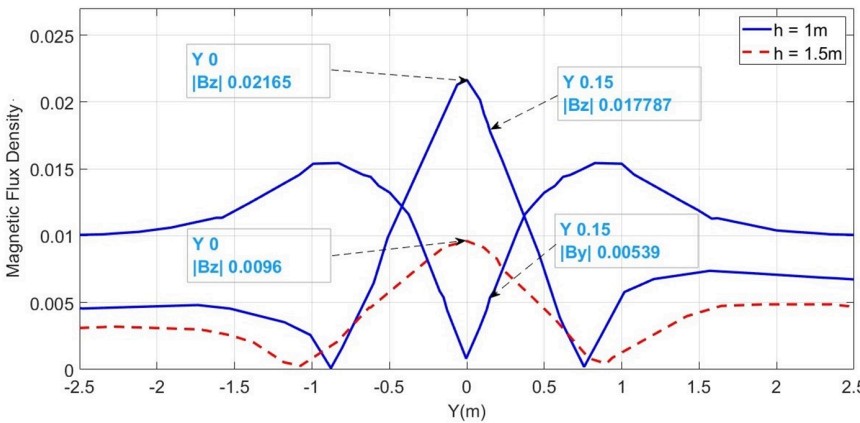

**Fig 16. $\vec{B}_y$ and $\vec{B}_z$ measured along line M for LLF at 100m.** The blue graphs represent magnetic flux densities when the cable is buried at h = 1m and the red dotted graph illustrates $\vec{B}_z$ when the cable is buried at h = 1.5m.

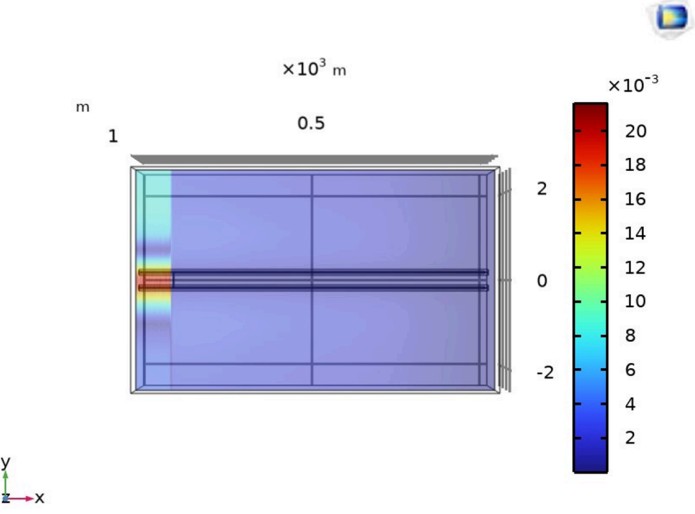

**Fig 17. Surface plot of $\vec{B}_z$ for LLF at 100m with the cables at a depth of h = 1m.**

Fig 16 with a red dotted graph. At y = 0m, the $\vec{B}_z$ is equal to $9.6 \times 10^{-3}$T, corresponding to a fault distance of 103.64m.

Additionally, the proposed method is applied to the $F_b$ and $F_c$ fault points, as illustrated in Fig 15. The $\vec{B}_z$ plots for the cables depths at h = 1m and h = 1.5m along line 'M' are presented in Figs 18 and 19, respectively. The average percentage errors are found to be 3.3% and 6.21 % for the $F_b$ and $F_c$ fault points, respectively. Table 4 presents the complete parameters and results of the LLF points and the credibility of the proposed method is demonstrated by a maximum error of 5.85% for the LLF.

In summary, the combined results of LGF, LCF, and LLF at cable depths of h = 1m and h = 1.5m are presented in Fig 20. The figure demonstrates the variation in percentage error at different fault locations along the cable. The analysis reveals that the maximum error observed is 7.78%, while the minimum error is 1.19%.

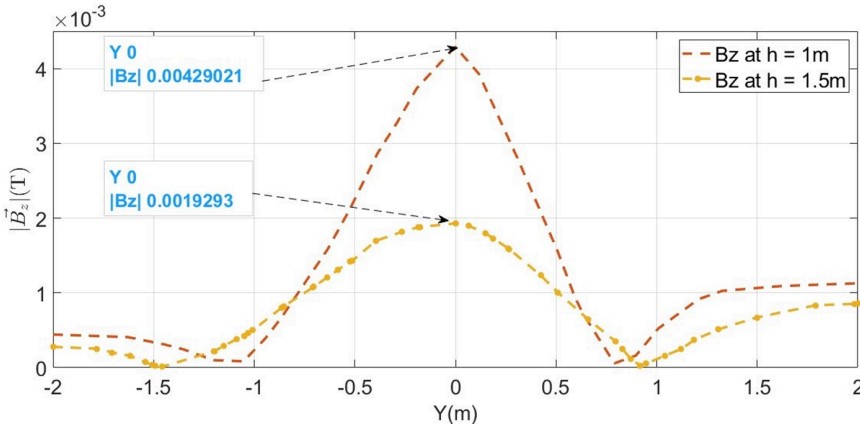

**Fig 18.** $\vec{B}_z$ measured along line M for LLF at 500m with cables buried at a depth of h = 1m and h = 1.5m.

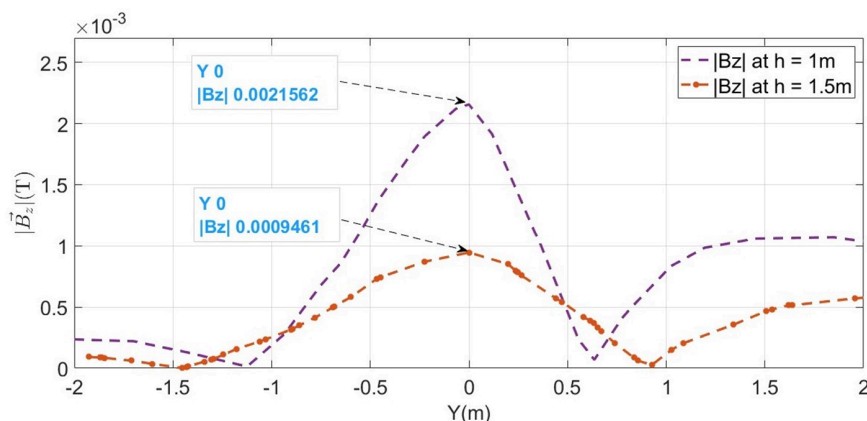

**Fig 19.** $\vec{B}_z$ measured along line M for LLF at 1000m with cables buried at a depth of h = 1m and h = 1.5m.

**Table 4.** Outcomes of line to line faults at 100m, 500m, and 1000m.

| $h$(m) | $|\vec{B}_z|$(mT) | $I_F$(kA) | $X_{act}$(m) | $X_{cal}$(m) | $E_\%$ |
|--------|-------------------|-----------|--------------|--------------|--------|
| 1      | 21.65             | 368.95    | 100          | 102.14       | 2.14   |
| 1.5    | 9.6               | 363.6     | 100          | 103.64       | 3.64   |
| 1      | 4.29              | 73.109    | 500          | 515.48       | 3.09   |
| 1.5    | 1.92              | 72.72     | 500          | 518.24       | 3.64   |
| 1      | 2.15              | 36.64     | 1000         | 1028.6       | 2.85   |
| 1.5    | 0.94              | 35.602    | 1000         | 1058.5       | 5.85   |

It is worth noting that the proposed method can locate multiple fault points sequentially on the same cable in the case of LGF and LLF, that is, after clearing the nearest fault point, subsequent fault points can be identified. However, simultaneous localization of multiple fault points on the same cable is not feasible with the current approach and remains a limitation for future work.

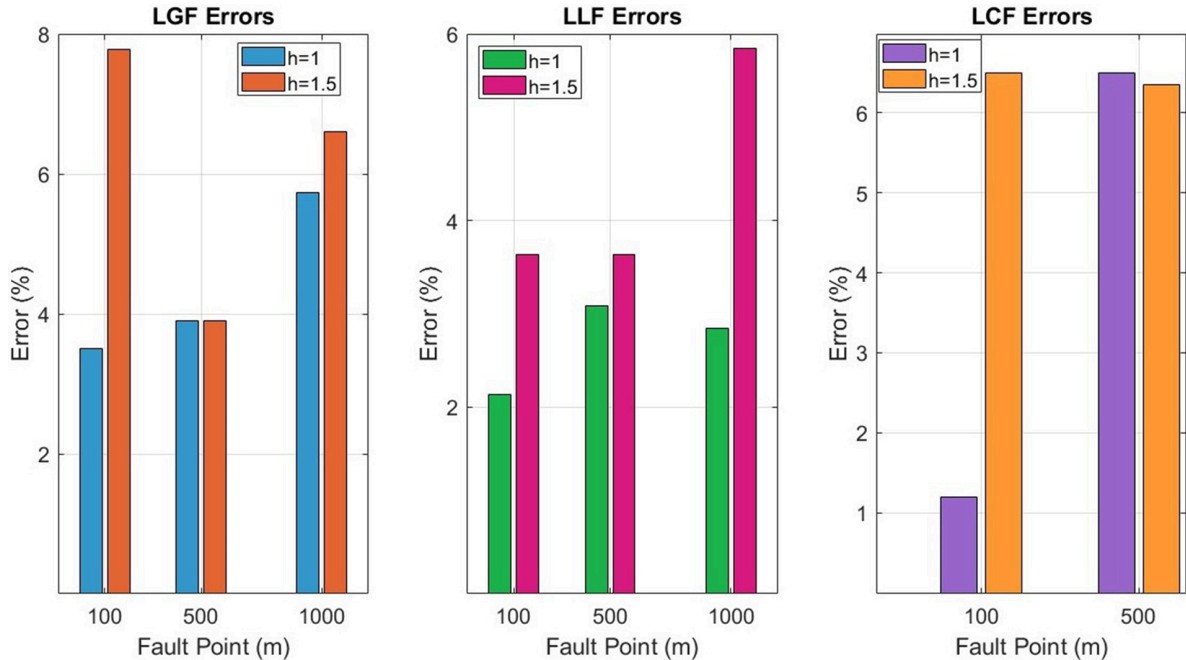

**Fig 20. Percentage error in the calculated fault distance for LGF, LLF, and LCF at cable depths of h = 1m and h = 1.5m.**

## Experimental validation

The feasibility of 2D fault localization is experimentally validated using the proposed single-axis magnetic field sensing method. Initially, the depth and the horizontal distance of LGF are calculated followed by the LLF.

## LGF

To verify the feasibility of the proposed method against the LGF, an experiment is conducted using a single-core copper cable of length 100 m, cross-sectional area $1.278 \times 10^{-6}$ mm$^2$, and resistivity approximately $8.6 \times 10^{-8}$ $\Omega$m . This cable is buried in soil inside a wooden box with dimensions of 1 ft $\times$ 1.5 ft, as illustrated in Fig 21. As shown in the figure, the single-core cable is stationed along the x-axis at a depth of 2 cm from the top of the wooden box and then later covered with soil. The cable is grounded at 90 m, thus creating an LGF with a grounding resistance of 3.07 $\Omega$. The ground resistance was measured at the earth pit installed in the campus yard using a Smart Sensor AR4105 earth resistance tester, which was subsequently connected to the LGF point. This ground resistance test using the earth resistance tester is illustrated in Fig 22.

Applying a 30 V DC across the cable using the Lab Volt GPS-3303 supply induces a static current in the cable. Fig 23 illustrates the experimental setup for the LGF localization. The magnetic flux density is sensed along line M using a Gauss meter GM09-3 [33], which has a measurement range of $\pm 1\,\mu$T to $\pm 2.5$ T. This magnetometer employs a three-axis (xyz) 3DHS-1 probe, which is moved along line M using a 3D numerical controller operated through its user interface, as illustrated in the figure. Along line M, 40 measurements were taken at 0.5 cm intervals, and the surrounding magnetic field is nullified by the null option $\varnothing$ in GM09-3.

The measured distributions of $|\vec{B_y}|$ and $|\vec{B_z}|$ along line M are shown in Fig 24. At point 18, $|\vec{B_y}|$ reaches its maximum value while $|\vec{B_z}|$ reaches its minimum value, confirming the location

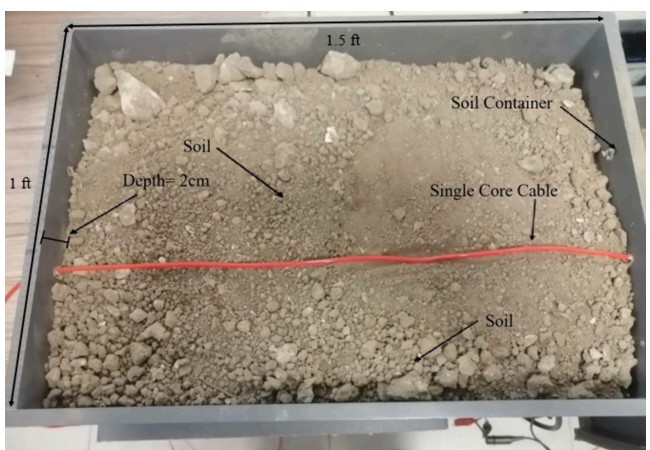

**Fig 21. Single-core cable buried along the x-axis at a depth of 2cm from the top of the wooden box.**

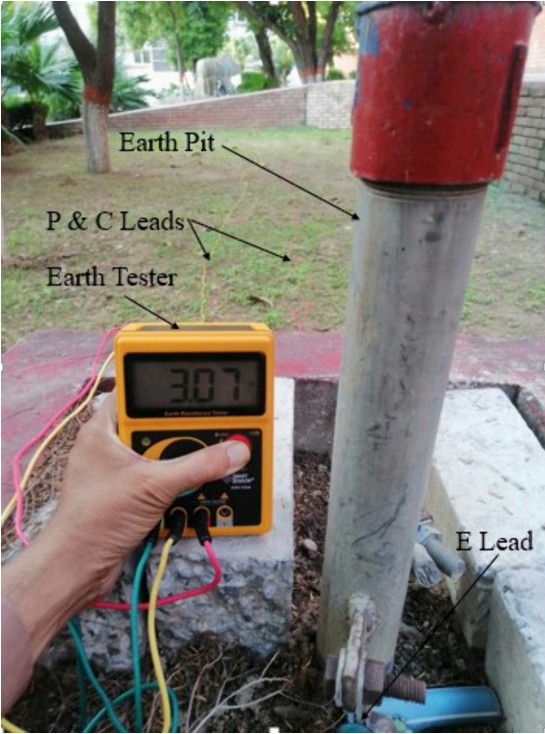

**Fig 22. Measurement of grounding resistance.** The potential (P) and current (C) leads are connected between the respective rods and the tester, while the earth (E) lead connects the tester to the earth pit.

of the cable on the surface. For the calculation of depth, at observation point 20, the measured values of $|\vec{B}_y|$ and $|\vec{B}_z|$ are 0.0232 mT and 0.012 mT, respectively. According to (9), the depth is calculated as 1.93 cm with a percentage error of 3.33%. Similarly, for the fault point, at cable's location, the $|\vec{B}_y|$ is measured as 0.03 mT. Using (8), the current is calculated as 2.9 A and with (2), the fault point is determined to be 108.1 m with a percentage error of 20.1%.

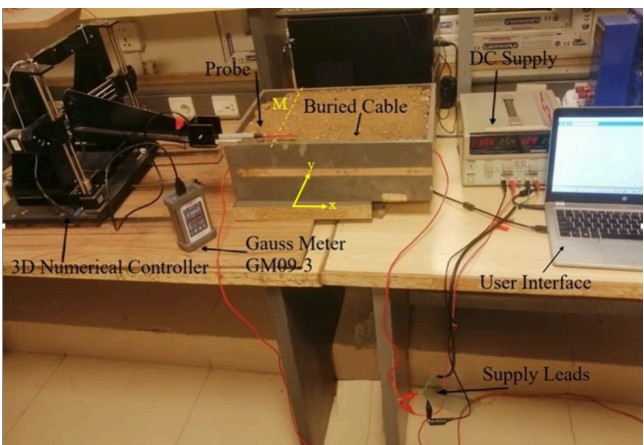

**Fig 23. Experimental setup for LGF localization.**

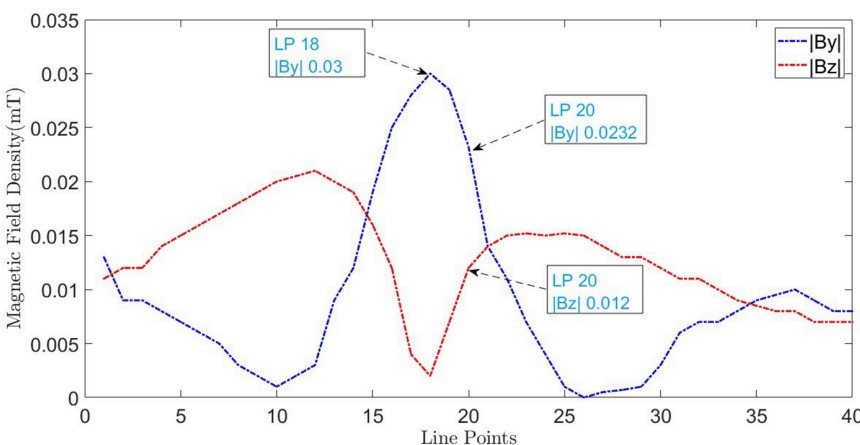

**Fig 24.** $\vec{B}_y$ **and** $\vec{B}_z$ **measured along line M for LGF at 90m.**

## LLF

The experimental setup for LLF localization using the proposed magnetic field sensing method is shown in Fig 25. The LLF point is created at 45 m by shorting two single-core cables (identical to the LGF cable) placed 9.5 cm apart and aligned along the x-axis. The LLF cables are buried at 2.3 cm in the same soil-filled box that was used for LGF localization.

To measure the magnetic flux density for LLF localization, a 30V DC is applied across the cables and the flux density is recorded along line M at the surface. Line M consists of 50 points spaced at intervals of 0.5 cm. The GM09-3 probe is moved along line M at the surface using the 3D controller and the measured magnetic flux densities $|\vec{B}_y|$ and $|\vec{B}_z|$ are shown in Fig 26. At point 26, $|\vec{B}_z|$ achieves its maximum value while $|\vec{B}_y|$ reaches its minimum value, confirming cable location. At point 44, $|\vec{B}_z|$ = 0.011 mT and $|\vec{B}_y|$ = 0.0026 mT. According to (21), the depth is calculated as 2.24 cm with an error of 2.3 %. Using (20), the resulting fault current is calculated to be 6.01 A. By applying (15), the fault distance is determined as x=31.52 m with an error of 29.9 %.

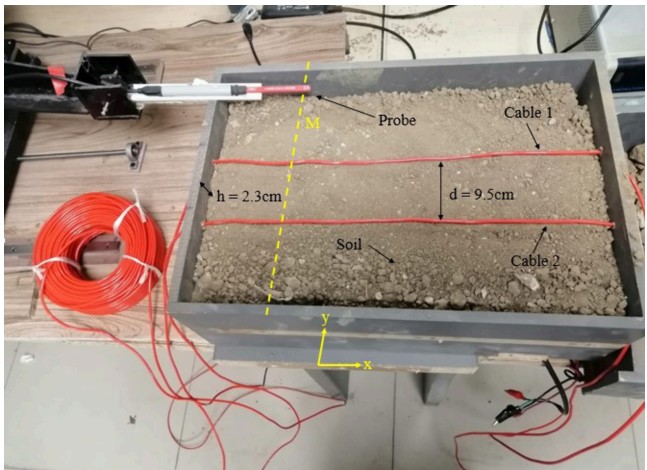

**Fig 25. Experimental setup for the LLF localization.**

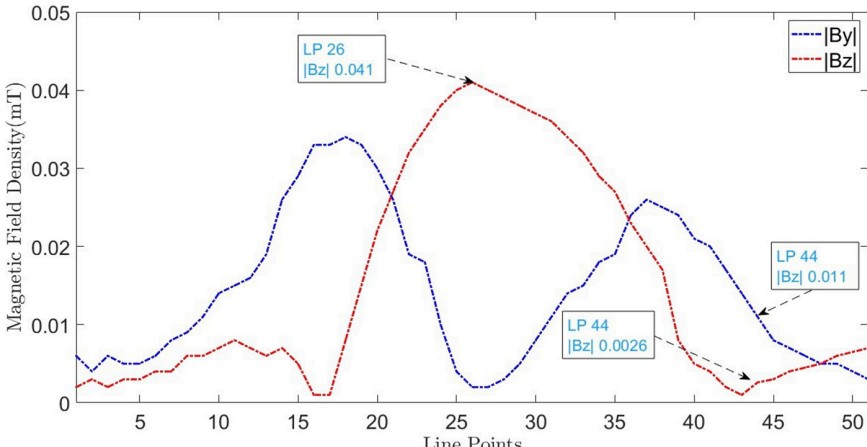

**Fig 26. $\vec{B}_y$ and $\vec{B}_z$ measured along line M for LLF at 45m.** The blue dotted graphs represent magnetic flux density of $\vec{B}_y$ and the red dotted graph illustrates $\vec{B}_z$ when the cable is buried at h = 2.3cm.

## Conclusion

A non-invasive magnetic field sensing method is proposed in this study for the localization of faults in underground cables. Unlike existing techniques, the proposed method determines both the cable's depth and the horizontal fault distance without requiring physical access to the cables. Moreover, the proposed method relies on a single movable magnetic sensor, in contrast to existing methods that depend on multiple fixed sensors, thereby making it cost effective. The effectiveness of the proposed method is validated through both simulations and experiments, demonstrating acceptable accuracy and highlighting its robustness across different fault types.

As future directions, we plan to test the proposed method in the field. While electromagnetic interference (EMI) can be mitigated using existing techniques, the development of more effective suppression strategies remains a subject for future research. Furthermore, the

deployment of drones for magnetic field sensing could make the proposed method even more effective by enabling remote fault localization, thereby reducing time and labor requirements.

## Author contributions

**Conceptualization:** Aamir Qamar.

**Funding acquisition:** Nayef Alqahtani, Ali H. Alenezi.

**Methodology:** Hamid Ali.

**Resources:** Nayef Alqahtani, Ali H. Alenezi.

**Software:** Hamid Ali.

**Supervision:** Aamir Qamar.

**Visualization:** Faheem Ali.

**Writing – original draft:** Hamid Ali.

**Writing – review & editing:** Aamir Qamar.

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
