## [Decision Letter · Decision Letter 0]

22 Jul 2025

PONE-D-25-36102Magnetic Field Sensing for 2D Fault Localization in Underground Power CablesPLOS ONE

Dear Dr. Qamar,

Thank you for submitting your manuscript to PLOS ONE. After careful consideration, we feel that it has merit but does not fully meet PLOS ONE’s publication criteria as it currently stands. Therefore, we invite you to submit a revised version of the manuscript that addresses the points raised during the review process. Please submit your revised manuscript by Sep 05 2025 11:59PM. If you will need more time than this to complete your revisions, please reply to this message or contact the journal office at plosone@plos.org. Please include the following items when submitting your revised manuscript:

We look forward to receiving your revised manuscript.

Kind regards,

Zeashan Hameed Khan, Ph.D.

Academic Editor

PLOS ONE

Journal Requirements:

“Deanship of Scientific Research at Northern Border University KSA, Grant Number: NBU-FER-2024-2159-04”

“Deanship of Scientific Research at Northern Border University KSA, Grant Number: NBU-FER-2024-2159-04”

“The author(s) received no specific funding for this work”

**Additional Editor Comments:**

This paper describes an approach based on magnetic field sensing for 2D fault localization in underground power cables. The conceptual framework and the implementation needs to be proven as novel and requires further enhancement. It is required to use more recent references for positioning your contribution well within the existing research.

Reviewers' comments:

Reviewer's Responses to Questions

**Comments to the Author**

1. Is the manuscript technically sound, and do the data support the conclusions?

Reviewer #1: Yes

Reviewer #2: No

Reviewer #3: Partly

2. Has the statistical analysis been performed appropriately and rigorously? 

Reviewer #1: N/A

Reviewer #2: No

Reviewer #3: Yes

3. Have the authors made all data underlying the findings in their manuscript fully available?

Reviewer #1: Yes

Reviewer #2: Yes

Reviewer #3: Yes

4. Is the manuscript presented in an intelligible fashion and written in standard English?

Reviewer #1: Yes

Reviewer #2: Yes

Reviewer #3: Yes

5. Review Comments to the Author

Reviewer #1: Comments to the Authors of the PONE-D-25-36102 manuscript are as follows:

1. The opinion of this reviewer is that this manuscript proposes a new method for locating various types of faults in underground power cables. However, none of the three faults considered has an experimental background that could be given for at least one of the three faults. Specifically, this study lacks an adequate experimental background. Finally, any new method must be validated using relevant experimental data.

2. There are similar methods in the literature that the Authors did not use in the literature review. Why? For example, the following publications: 10.1109/TMAG.2013.2297195, 10.1061/41202(423)242, etc.

3. Novelties and scientific contributions must be contrasted to the relevant state-of-the-art and highlighted in an appropriate manner.

4. A reference should be provided in the list of references for the version of COMSOL used.

5. The accuracy of the FEM-based models used must be verified by means of the results of appropriate FE mesh independence tests. Without these tests the FEM-based model can be regarded as incomplete. How to conduct these tests can be found in some papers recently published.

6. The following sentence: "This leakage current can cause insulation failure and could become a major fault in the near future." is not based on research results and represents the Authors' opinion and should be modified accordingly.

7. Some abbreviations are defined more than once.

8. The grounding resistance is said to be assumed. Specifically, it says: "The grounding resistance is assumed to be 0.294534 Ohm". How was this value selected or estimated?

9. The conclusions must be improved in terms of scientific significance and quantified additionally based on the obtained results.

Reviewer #2: The paper explains three types of faults in Underground Power Cables (UPC): LGF, LCF, and LLF, and proposes an inverse solution algorithm based on Kirchhoff’s Voltage Law (KVL) for fault location. However, the paper lacks comparative experiments, making it difficult to assess the proposed method's effectiveness against existing techniques. The innovation of the method is limited, and the paper does not sufficiently highlight how it improves upon current approaches. Additionally, the writing quality and structure could be improved, particularly in the explanation of the methodology and results. The algorithm is validated only through simulations, with no real-world experimental verification, and there is insufficient explanation of the magnetic field sensing equipment used. Therefore, the paper is not recommended for publication. The detailed comments are as follow:

1、The magnetic field sensing equipment mentioned in the abstract is only validated through simulations in this paper, with no experimental verification using actual devices. Additionally, there is insufficient explanation of the equipment used in the study.

2、The use of abbreviations throughout the paper is inconsistent and confusing.

3、The method proposed in this paper lacks comparative experiments, making it difficult to directly demonstrate the algorithm’s time complexity, power consumption, and the high cost associated with the large number of devices required.

4、The line charts used in Figures 20 and 21 are not suitable for effectively representing the specific errors. It is recommended to replace them with other types of graphs for better clarity.

5、The fault points described in the paper are at 100m, 500m, and 1000m. If all three points were to fail simultaneously, it is unclear whether the proposed method can accurately locate the faults. Further clarification or testing on this scenario would be necessary.

6、In the paper, the tables present the values of Xact and Xcal along with their corresponding errors. However, the percentage error does not effectively reflect the precision of the proposed method. Additionally, there is a lack of comparative algorithms and their results, making it difficult to assess the effectiveness of the proposed algorithm.

Reviewer #3: This paper presented a magnetic field sensing for Fault Location in underground cables. The research is interesting. However, there are some concerns raised by this reviewer. General comments are as follows.

(1) In introduction part, the authors stated that the proposed method is cost effective. However, there is no verification on this statement.

(2) The paper is heavy on theoretical analysis but short on practice. How to measure the magnetic field? What kind of sensor is applicable to achieve magnetic field sensing for fault location? The impacts of environmental interferences (noises, electromagnetic, sampling interferences, etc.) should be considered.

(3) What are the differences between LGF and LCF? The detailed fault developing mechanisms for the two types of faults are required.

(4) In (2) and (13), the fault resistance Rg commonly vary with fault conditions. The reason why it is set as a constant should be explained.

(5) Why magnetic field is used? Why not measure current or voltage signals? There are many sensors for electrical signals in practice but few sensors for field quantities. A comparison work is needed.

(6) COMSOL is used to conduct the simulations. However, it is just an offline simulation tool. The reason why not use field data should be explained.

6. PLOS authors have the option to publish the peer review history of their article (what does this mean?). If published, this will include your full peer review and any attached files.

Reviewer #1: No

Reviewer #2: No

Reviewer #3: No

---

## [Author Response · Author response to Decision Letter 1]

15 Sep 2025

The response letter is attached to this online submission.

---

## [Decision Letter · Decision Letter 1]

22 Sep 2025

PONE-D-25-36102R1Magnetic Field Sensing for 2D Fault Localization in Underground Power CablesPLOS ONE

Dear Dr. Qamar,

Thank you for submitting your manuscript to PLOS ONE. After careful consideration, we feel that it has merit but does not fully meet PLOS ONE’s publication criteria as it currently stands. Therefore, we invite you to submit a revised version of the manuscript that addresses the points raised during the review process.

We look forward to receiving your revised manuscript.

Kind regards,

Zeashan Hameed Khan, Ph.D.

Academic Editor

PLOS ONE

Journal Requirements:

Additional Editor Comments:

The revised version has been improved to include all comments of the reviewers. However, some minor improvements are still needed. Please have a careful look at the manuscript to respond to the remaining queries.

Reviewers' comments:

Reviewer's Responses to Questions

**Comments to the Author**

1. If the authors have adequately addressed your comments raised in a previous round of review and you feel that this manuscript is now acceptable for publication, you may indicate that here to bypass the “Comments to the Author” section, enter your conflict of interest statement in the “Confidential to Editor” section, and submit your "Accept" recommendation.

Reviewer #1: (No Response)

Reviewer #3: (No Response)

2. Is the manuscript technically sound, and do the data support the conclusions?

Reviewer #1: Yes

Reviewer #3: (No Response)

3. Has the statistical analysis been performed appropriately and rigorously? 

Reviewer #1: N/A

Reviewer #3: (No Response)

4. Have the authors made all data underlying the findings in their manuscript fully available?

Reviewer #1: Yes

Reviewer #3: (No Response)

5. Is the manuscript presented in an intelligible fashion and written in standard English?

Reviewer #1: Yes

Reviewer #3: (No Response)

6. Review Comments to the Author

Reviewer #1: 1. The following sentence "The experimental setup for LLF localization using the proposed magnetic field sending

method is shown in Fig. 25." should be corrected.

2. The results of FE mesh independence tests do not verify the accuracy of the proposed method. Authors are requested to perform these tests using papers published in the following journals: Applied Thermal Engineering, Electrical Engineering, Thermal Science, etc.

3. In the following sentence "In these studies, the magnetic flux density of live cables was utilized to determine both the horizontal position and the vertical depth of the cable.", the term "live cables" should be replaced with any other more convenient term.

Reviewer #3: (No Response)

7. PLOS authors have the option to publish the peer review history of their article (what does this mean?). If published, this will include your full peer review and any attached files.

Reviewer #1: No

Reviewer #3: No

---

## [Author Response · Author response to Decision Letter 2]

25 Sep 2025

Response letter is attached to the this submission.

---

## [Decision Letter · Decision Letter 2]

30 Sep 2025

Magnetic Field Sensing for 2D Fault Localization in Underground Power Cables

PONE-D-25-36102R2

Dear Dr. Qamar,

We’re pleased to inform you that your manuscript has been judged scientifically suitable for publication and will be formally accepted for publication once it meets all outstanding technical requirements.

Kind regards,

Zeashan Hameed Khan, Ph.D.

Academic Editor

PLOS ONE

Additional Editor Comments (optional):

After the second review cycle, the authors have addressed all the major discrepancies and the paper is acceptable in the present form.

Reviewers' comments:

Reviewer's Responses to Questions

**Comments to the Author**

1. If the authors have adequately addressed your comments raised in a previous round of review and you feel that this manuscript is now acceptable for publication, you may indicate that here to bypass the “Comments to the Author” section, enter your conflict of interest statement in the “Confidential to Editor” section, and submit your "Accept" recommendation.

Reviewer #1: All comments have been addressed

Reviewer #3: All comments have been addressed

2. Is the manuscript technically sound, and do the data support the conclusions?

Reviewer #1: Yes

Reviewer #3: (No Response)

3. Has the statistical analysis been performed appropriately and rigorously? 

Reviewer #1: N/A

Reviewer #3: (No Response)

4. Have the authors made all data underlying the findings in their manuscript fully available?

Reviewer #1: Yes

Reviewer #3: (No Response)

5. Is the manuscript presented in an intelligible fashion and written in standard English?

Reviewer #1: Yes

Reviewer #3: (No Response)

6. Review Comments to the Author

Reviewer #1: There are no additional comments for the Authors. All the comments from the previous round of review were addressed as requested by this reviewer.

Reviewer #3: (No Response)

7. PLOS authors have the option to publish the peer review history of their article (what does this mean?). If published, this will include your full peer review and any attached files.

Reviewer #1: No

Reviewer #3: No

---

## [Editor Report · Acceptance letter]

PONE-D-25-36102R2

PLOS ONE

Dear Dr. Qamar,

I'm pleased to inform you that your manuscript has been deemed suitable for publication in PLOS ONE. Congratulations! Your manuscript is now being handed over to our production team.

Kind regards,

on behalf of

Dr. Zeashan Hameed Khan

Academic Editor

PLOS ONE